# Full-Length Transcriptome Sequencing and Comparative Transcriptome Analysis to Evaluate Drought and Salt Stress in *Iris lactea* var. *chinensis*

**DOI:** 10.3390/genes12030434

**Published:** 2021-03-18

**Authors:** Longjie Ni, Zhiquan Wang, Jinbo Guo, Xiaoxiao Pei, Liangqin Liu, Huogen Li, Haiyan Yuan, Chunsun Gu

**Affiliations:** 1Institute of Botany, Jiangsu Province and Chinese Academy of Sciences, Nanjing 210014, China; LongJieNi@njfu.edu.cn (L.N.); wangzhiquan@cnbg.net (Z.W.); guojinbo@cnbg.net (J.G.); peixiaoxiao@cnbg.net (X.P.); liuliangqin@cnbg.net (L.L.); yuanhaiyan@cnbg.net (H.Y.); 2College of Forest Sciences, Nanjing Forestry University, Nanjing 210037, China; hgli@njfu.edu.cn; 3Jiangsu Provincial Platform for Conservation and Utilization of Agricultural Germplasm, Nanjing 210014, China

**Keywords:** *Iris lactea* var. *chinensis*, drought-stress tolerance, salt-stress tolerance, full-length transcripts, differentially expressed genes

## Abstract

*Iris lactea* var. *chinensis* (*I. lactea* var. *chinensis*) is a perennial herb halophyte with salt and drought tolerance. In this study, full-length transcripts of *I. lactea* var. *chinensis* were sequenced using the PacBio RSII sequencing platform. Moreover, the transcriptome was investigated under NaCl or polyethylene glycol (PEG) stress. Approximately 30.89 G subreads were generated and 31,195 unigenes were obtained by clustering the same isoforms by the PacBio RSII platform. A total of 15,466 differentially expressed genes (DEGs) were obtained under the two stresses using the Illumina platform. Among them, 9266 and 8390 DEGs were obtained under high concentrations of NaCl and PEG, respectively. In total, 3897 DEGs with the same expression pattern under the two stresses were obtained. The transcriptome expression profiles of *I. lactea* var. *chinensis* under NaCl or PEG stress obtained in this study may provide a resource for the same and different response mechanisms against different types of abiotic stress. Furthermore, the stress-related genes found in this study can provide data for future molecular breeding.

## 1. Introduction

Plants are sessile organisms and are often affected by various abiotic stresses during their life cycle. These abiotic stresses include drought, salt damage, and toxic metals in soil, among which salt damage and drought stress are important factors limiting plant growth and productivity [1]. Moreover, with the increasingly frequent occurrence of extreme weather, the semi-arid and arid areas of the world continue to increase, forcing agriculture to gradually expand to semi-arid or arid areas. In addition, intensive agricultural irrigation will lead to changes in soil water balance in these areas and aggravate soil salinization [2]. Therefore, it is critical to increase plant resistance to drought and salt for agricultural production and the sustainable use of the environment. At the same time, the water and fertilizer consumption of stress-resistant plants will also be greatly reduced, thus reducing the cost and environmental burden.

Drought and salt can cause plants to have high permeability in the early stages of stress. At the same time, salt stress can cause an ionic imbalance. Over time, increased drought and salt stress will gradually generate oxidative stress and cause damage to cell components, such as membrane lipids, and proteins, and metabolic disorders caused by a series of complex secondary stress. Therefore, in response to drought and salt stress, plants have separate and common signal transduction mechanisms [3]. The most important feature of drought and salt stress is that it can promote the accumulation of the plant hormone abscisic acid (ABA), which leads to a series of adaptive responses in plants [4]. In addition, the increase of Ca^2+^ and reactive oxygen species (ROS) are key signal transduction components in the plant response to these two stresses [5]. When plants accumulate levels of Ca^2+^, ROS, and ABA, they will further regulate downstream drought and salt stress-related transcription factors (TFs) through signaling cascade reactions, thus further changing the transcription level in plants and improving stress resistance [6]. Current research on the mechanisms of drought and salt stress regulation has made great progress, for example, by finding the salt overly sensitive (SOS) pathway [7], ABA signal transduction pathway [8], and salt ion receptors glycosyl inositol phosphorylceramide (GIPC) sphingolipids [9]. However, more research about the underlying mechanisms of plant response to drought and salt stress is needed, especially in non-model plants.

*I. lactea* var. *chinensis* is a Chinese perennial herbaceous halophyte with wide adaptability and ornamental value. It is an important plant for ecological construction and the improvement of the soil environment [10]. Next-generation sequencing (NGS) technology has been used to reveal the regulatory mechanism of *I. lactea* var. *chinensis* under salt stress [11]. To our knowledge, however, there are no comparative studies on the common regulation mechanism of *I. lactea* var. *chinensis* under drought and salt stress. Moreover, due to the obvious disadvantages of second-generation sequencing, such as the short length of sequencing reads and unreliable assembly results, the accuracy of transcriptome abundance calculation is reduced to a large extent [12,13].

PacBio third-generation sequencing technology can effectively overcome problems associated with NGS. Longer read length, more uniform coverage, and more other advantages in building a complete transcriptome can be obtained by Pacbio, but at the same time, non-assembled long-read transcripts with a low error rate (10%) can be generated by Pacbio, and the error rate can be overcome with correction of Illumina RNA-Seq [14]. Currently, these two sequencing technologies have been used to study different regulatory mechanisms in multiple species. For example, more transcriptional splice events have been found in rice by combining these two technologies [15], and they have both been used to reveal the complexity of ginsenoside biosynthesis and the transcriptome in *Panax notoginseng* [16]. Thus, for the first time, we constructed full-length reference transcriptomes of *I. lactea* var. *chinensis* under PEG and NaCl stress using isoform sequencing (ISO-seq sequencing) technology. According to the RNA-Seq data obtained by the Illumina sequencing platform, expression levels were compared and analyzed. The results revealed the changes of transcripts in *I. lactea* var. *chinensis* under 24 h PEG and NaCl stress at the molecular level. These results will offer new data for the future study of the mechanism involved in the regulation of abiotic stress in *I. lactea* var. *chinensis*.

## 2. Materials and Methods

### 2.1. Plant Materials, Growth Conditions, and Stress Treatment

The plant materials of *I. lactea* var. *chinensis*, which were conserved in the Iris Resource Collection Garden of the Nanjing Sun Yat-Sen Memorial Botanical Garden (Nanjing, China), were used in this research. The seeds were germinated on moist filter papers for 24 h and then cultivated under a sterile condition in a greenhouse for three weeks (16 h/8 h light/dark; 65% relative humidity). Afterward, the seedlings with similar tap root lengths were transferred into 50 mL centrifuge tubes and then cultured in half-strength Murashige and Skoog (1/2 MS) nutrient solution at pH 5.8 for one week. The solution was refreshed every two days during the seedling growth. Then the seedlings were set into 1/2 MS containing 1.5% NaCl or 30% PEG for 24 h, and the control seedlings were still cultured in standard 1/2 MS solution. The control and treatments were repeated three times with a total of nine biological replicates, and all samples (leaves and roots (approximately 1.5 cm in length)) were sampled at the same time for sequencing analysis; the samples were frozen in liquid nitrogen and deposited at −80 °C.

### 2.2. Iso-Seq Library RNA Preparation, Sequencing, and Analysis

The RNAiso reagent (TaKaRa Biotech Co., Dalian, China) was used to extract total RNA from each sample. A 1% agarose gel was used to monitor RNA degradation and contamination. Nanodrop 2000 spectrophotometer (Thermo Fisher Scientific Inc., Walthman, MA, USA) and Agilent 2100 Bioanalyzer (Agilent Technologies, Palo Alto, CA, USA) were used to monitor protein contamination (A260/A280 ratio) and reagent contamination (A260/A230 ratio) to validate RNA accuracy. The total RNA with ratios of A260/A280 and A260/A230 between 1.8 and 2.0 were chosen for subsequent analysis. The Qubit 2.0 fluorometer (Invitrogen, Carlsbad, CA, USA) was used to accurately quantify the RNA concentration of all samples.

The concentration of RNA of each sample was greater than 300 ng/ul, and the total amount of the RNA used for cDNA library was greater than 5 ug. The SMARTer^®^ PCR cDNA Synthesis Kit (Clontech, Mountain View, CA, USA) was used to synthesize the first-strand cDNA, and then second-strand cDNA was synthesized by large-scale PCR. A Kapa Hifi PCR package (KAPA Biosystems, Boston, MA, USA) was used for PCR amplification and cDNA synthesis. Bluepippin was used to screen fragments (optional size > 4 kb) and a SMRTBell template Planning kit 1.0 (Clontech, CA, USA) was used to build the SMRTBell library for damage and final repair. Finally, the stem-loop sequencing adapter was attached to both ends of the DNA fragments and exonuclease was used to delete the failed fragments. After the library was quantified, the library prototype enzyme complex was sequenced on a PacBio sequel system. The PacBio RSII tool (Pacific Bio-science Inc., Menlo Park, CA, USA) was used to sequence the cDNA library and then process polymerase readings to extract low-quality and short-read sequences. High-quality full-length sequences were obtained by grouping, clustering, and correction. SMRTlink 5.1 (http://www.pacb.com/products-and-services/analytical-sofware/smrt-analysis/, accessed on 1 August 2017) software was carried out for processing the above data, and the parameters are set as follows: min_length 300, max_drop_fraction 0.8, no_polish TRUE, min_zscore −9999, min_passes 1, min_predicted_accuracy 0.8, max_length 15,000.

CD-HIT (version: 4.6.7) was used to cluster the corrected transcript sequences according to the 95% similarity between the sequences, and the parameters are set as follows: -c 0.85, -T 6, -G 0, -aL 0.00, -aS 0.99, -AS 30 [17]. The core conserved gene set of terrestrial plants, namely, Eukaryota (version: V1, number of BUSCOs: 429), and BUSCO (version: 3.0.2) were used to evaluate the completeness of the full-length transcriptome sequences [18]. The TransDecoder program was used to estimate the ORF of the transcripts (version: 5.5.0; parameter: -m 50). The identification parameters were as follows: (1) open reading frame (ORF) length is greater than 300 bp and (2) logarithm value of the probability function is greater than 0.

All full-length transcripts were annotated using seven online libraries, including the NCBI non-redundant protein (NR) database, the Cluster of Orthologous Groups of proteins (COG), UniProt Knowledgebase (UniProt), the Kyoto Encyclopedia of Genes and Genomes (KEGG) database, the EuKaryotic Orthologous Groups (KOG) database, the Gene Ontology (GO) database, and the Protein Family (Pfam) database. The first five annotation databases were performed using DIAMOND (version: 0.8.36) with an E-value threshold of 1.0 × 10^−5^. HMMER 3.1 package was used for Pfam database annotation. Blast2GO v5.1 (http://www.blast2go.com, accessed on 1 May 2018) and a script were used for GO annotation.

### 2.3. RNA-Seq Library RNA Preparation, Sequencing, and Analysis

The amount of RNA of each sample was 1.5 ug for RNA-Seq library construction. NEBNext^®^Ultra™ RNA Library Prep Kit for Illumina^®^ (NEB, USA) was used to generate an RNA-Seq library following the manufacturer’s protocol. AMPure XP system (Beckman Coulter, Beverly, CA, USA) was used to select cDNA fragments of preferentially 250~300 bp in length and purify PCR products. Illumina Hiseq2000 (Illumina Inc., San Diego, CA, USA) was used for library sequencing and generating paired-end reads.

A full-length non-chimeric transcript was used as a reference sequence to obtain isoform after removing redundancy with CD-HIT, and Bowtie2 (version 2.3.4; Parameter: Mismatch 0) software was used to compare the second-generation high-throughput sequencing data with the above reference sequence. RSEM (version: 1.3.1) was used to obtain the fragments per kilobase of exon model per million mapped fragments (FPKM) based on the read count for each gene in each sample, and the FPKM value was converted to transcript per million (TPM) value to analyze the expression level [19].

### 2.4. DEGs Analysis

The edgeR was used to screen DEGs based on read counts, FDR < 0.05 and |log_2_(Foldchange)| > 1 were set as the threshold [20]. Then DEGs was also assessed via the DEseq2 R package, qvalue < 0.05 and |log_2_ (Foldchange)| > 1 were set as the threshold [21,22]. Finally, the DEGs, which were screened by both of the two soft wares, were used for subsequent analysis.

For function annotation, we used the GOseq R packages based on Walleniusnon-central hypergeometric distribution to perform GO enrichment analysis [23]. The KEGG enrichment analysis was carried out using KOBAS software [24], Venn diagrams and heat maps were drawn by TBTools (version: 1.007) [25], and the gene expression map was drawn by MapMan (version: 3.5.1) [26]. Coexpression networks were constructed and used with the WGCNA R packages (version: 1.68) [27], and visualization was performed with Cytoscape (version 1.7) [28]. The Cytoscape plugin cytoHubba was used to identify hub genes and rank them according to the maximal clique centrality (MCC) scores [29].

## 3. Results

### 3.1. Single-Molecule Real-Time (SMRT) Sequencing, Data Processing, Annotation, and Coding Sequence (CDS) Prediction

Total RNA was extracted from the root, stem, and leaf tissues of nine samples, including 24 h of NaCl treatment, 24 h of PEG treatment, and controls without any treatment. Almost equal amounts of high-quality RNA were mixed to generate an informative reference transcript database. Full-length transcriptome data were obtained by the PacBio Sequel platform, and approximately 30.89 G subreads (258,615 circular consensus sequencing (CCS)) were generated. After filtering out incomplete CGs, 168,924 full-length non-chimeric read (FLNC) sequences with complete 5′–3′ ends were obtained. After clustering redundant sequences, 99,483 consensus sequences were obtained, and then 62,717 isoforms were obtained by CD-HIT software (Figure 1a). Finally, 31,195 unigenes were obtained by clustering the same isoforms (Figure 1b), and the length of these sequences ranged from 161 to 6386 bp. The average size was 1564 bp, and the length of ExN50 was 1115 bp. A total of 23,244 (74.51%) transcripts were over 1000 bp in length. Based on benchmarking universal single-copy ortholog (BUSCO) analysis, approximately 307 (72%) of the 429 expected embryophyte genes were identified as complete (Appendix A
Appendix A). A total of 30,295 CDS sequences were predicted by TransDecoder software, among which 327 (1.07%) CDS sequences were longer than 3000 bp, 24,167 (79.75%) were between 400 bp and 3000 bp, and 5801 (19.15%) were less than 400 bp (Table 1), the remaining 900 genes without CDS may be pseudogenes and lncRNAs [30].

In this study, 31,195 unigenes were annotated using seven databases. The number of annotated unigenes in the seven databases ranged from 8633 (27.67%, KEGG pathways) to 28,063 (89.96%, NR), and 28,419 (91.10%) unigenes were annotated in at least one database (Appendix A). In addition, 21,697 unigenes were annotated to the GO database (Figure 1c) and 12,556 unigenes were annotated to the KEGG database (Figure 1d). A large number of genes were annotated with COG to the RNA processing and modification, followed by signal transduction mechanisms, defense mechanisms, and energy production andconversion (Figure 1f).

### 3.2. Illumina Sequencing Data Analysis and DEGs Screening

Using the Illumina platform, nine cDNA libraries (three replicates per process) were sequenced. A total of 431,683,966 reads were generated, with an average of 47,964,885 reads per library. Then, all reads were mapped to the full-length transcriptome database of *I. lactea* var. *chinensis* through the Bowtie2 software. The final alignment was 215,841,983 reads, and the average alignment was 23,250 (74.53%) unigenes. A total of 62,718 transcripts were identified in the nine sample libraries. Each library contained more than 20,000 genes. The average mapping rate of all libraries of RNA-seq in this study was 75.9%, suggesting that the library of PacBio had a high degree of integrity (Appendix A).

Based on the reads counts, DEGs were assessed with the DESeq2 program. A total of 15,466 DEGs were found to participate in response to the two stresses. Among them, compared with the CK, 9266 (5148 up-regulated, 4118 down-regulated) and 8390 (4827 up-regulated, 3563 down-regulated) DEGs were obtained under NaCl and PEG treatment, respectively (Figure 2c–f).

### 3.3. Analysis of Stress Regulation Networks

To screen the key regulatory genes under the two stresses further, weighted gene coexpression network analyses (WGCNA) of all DEGs were performed, and DEGs were divided into 12 different hierarchical clustering modules. Different colors (blue, darkgreen, grey, darkgrey, darkred, green, lightgreen, paleturquoise, red, royalblue, steelblue, and white) were used to represent different modules, and GO enrichment analysis was carried out for different modules (Figure 3a,b).

The largest module, darkred, contained 5053 DEGs, most of which were enriched in the membrane, transmembrane transporter activity, plasma membrane, and other GO terms related to the transporter, and showed the strongest positive relationship with the control group. The second module, blue, contained 3447 DEGs. Most of the genes in this module were enriched in response to stimuli, protein modification process, and other GO terms closely related to metabolism, and showed the strongest positive relationship with PEG stress. The third module, darkgrey, contained 2192 DEGs. Most of the genes in this module were enriched in catalytic activity, cell wall, membrane, and other GO terms related to cell structure, and showed the strongest positive relationship with NaCl stress (Figure 3c).

Cytoscape Plugin Cytohubba was used to selected 10 DEGs, as the key genes, from the darkgrey and blue modules and were annotated with NCBI blast (Figure 3d,e). The key genes under salt stress were annotated as calcium permanent channel 1 (*OSCA1*), probable protein phosphatase 2C (*PP2C*), mitogen-activated protein kinase (*MAPK*), calcium-dependent protein kinase 19 (*CDPK*), abscisic acid receptor PYL10 (*PYL10*), dihydroflavonol 4-reductase (*DFRA*), chalcone synthase 1 (*CHS1*), ATP-citrate synthase β chain protein 1 (*ACLB1*), calcium-dependent protein kinase 5 (*CDPK5*), and acetyl-CoA carboxylase 1(*ACC1*) (Appendix A), and the key genes under drought stress were annotated as phosphoenolpyruvate carboxylase kinase 1(*PPCK1*), *PP2C*, succinate, probable WRKY transcription factor 15 (*WRKY15*), calcineurin B-like (CBL) interacting protein kinase (*CIPK8*), stem-specific protein (*TSJT1*), glutathione S-transferase (*GST*), serine/threonine-protein kinase (*STY17*), and translationally controlled tumor protein homolog (Appendix A).

### 3.4. Analysis of Comparative Transcriptome

A total of 15,466 DEGs were found participating in response to the two stresses (Figure 4a). All the DEGs were divided into four categories according to their expression patterns under different stresses. Among them, those that were upregulated under NaCl and PEG stress were divided into categories I and III, and those that were downregulated under NaCl and PEG stress were divided into categories II and IV. GO enrichment analysis showed that the DEGs under NaCl and PEG stress have similar functions. Among them, transporter activity, transmembrane transporter activity, cell wall, cell periphery, and membrane were significantly enriched under the two stresses. However, only type I, type II, and type IV genes were significantly enriched in these terms, and not type III. This suggested that the same genes may play different roles in response to these two stresses. In addition, transmembrane transport, small molecular metabolic processes, and oxidoreductase activity were only enriched in type I and type III, indicating that these terms may play a positive regulatory role in response to the two stresses. On the other hand, carbon metabolic processes, plasma membrane, and vacuum were only enriched in type II and type IV, indicating that these terms may play a negative regulatory role in response to the two stresses. In addition, transport, lipid metabolic process, peptidase activity, and catalytic activity were only enriched under NaCl stress, whereas DNA binding, Golgi attachments, nuclear acid binding transcription factor activity, and response to stimulus were only enriched in PEG stress, indicating that *I. lactea* var. *chinensis* had different defense mechanisms in response to the two stresses (Figure 4b).

To explore the common defense mechanism of *I. lactea* var. *chinensis* in response to the two stresses further, we analyzed DEGs with the same expression pattern under the two stresses separately. In this study, 3863 DEGs had the same expression pattern under the two stresses, and the remaining 34 DEGs had different expression patterns under the two stresses (Figure 4a). Among them, 2508 DEGs were upregulated and 1355 DEGs were downregulated. GO enrichment analysis showed that the two types of DEGs were enriched in five GO terms, namely transporter activity, transmembrane transporter activity, external covering structure, cell wall, and catabolic process, which were closely related to stress response, indicating that these GO terms may play important roles in response to stress. In addition, upregulated DEGs were specifically enriched to nine GO terms, including oxidoreductase activity, response to stress, single-organization transport, and transmembrane transport, indicating that these terms may play positive regulatory roles in response to stress, whereas down-regulated DEGs were specifically enriched to seven GO terms, including the plasma membrane, membrane, cell periphery, and carbohydrate metabolic process, suggesting that these terms might play a negative regulatory role in response to stress (Figure 4c). KEGG enrichment analysis was used to understand the complex biological behavior. Overall, 3863 DEGs with the same expression pattern were enriched in 17 pathways, including amino acid metabolism, carbon hydrate metabolism, signaling, and cellular processes, ion channels, transporters, and other pathways related to metabolism and ion transport (Figure 4d). This suggested that the defense of *I. lactea* var. *chinensis* in response to these two stresses was through metabolism and ion transport pathways.

Osmotic stress is the core of the plant response to drought and salt stress; hence, we mapped the related DEGs to the osmotic stress regulatory network (Figure 5).

The results showed that a total of 22 DEGs were mapped to the ABA signaling pathway, of which six DEGs were mapped to the ABA receptor *PYR*s/*PYL*, and all of them were in a downward trend; five DEGs were mapped to the threonine/serine receptor kinase *SnRK2*, and all of them were in an upward trend; seven DEGs were mapped to the ABA coreceptor *PP2C*, most of them were in an upward trend; and one DEG mapped to rapid alkalinization factor 1 (RALF), two mapped to receptor-like kinase FERONIA (FER), and 1 mapped to the guanine nuclear exchange factor (GEF) were all in a downward trend.

A total of 35 DEGs were mapped to the osmotic stress signaling pathway, among which *MAPK*s were mapped to six and three were upregulated; secondary messengers *CPK*s and *CBL*s-*CIPK*s were mapped to 18 and 7 were upregulated; potassium channel (*KAT1*) was mapped to one DEG, which was upregulated.

In addition, 85 DEGs were mapped as TFs. Among them, the expression of 1 *ABF* was upregulated; 17 *WRKY*s were all upregulated; three of six *bHlH*s were upregulated; 23 *NAC*s were all upregulated; 6 of 11 *MYB*s were upregulated; 12 of 18 *AP2/ERF*s were upregulated; and nine *GRAS*s were all upregulated. It is worth noting that *POD*, *CAT*, and *GST* related to non-enzymatic antioxidants have also been mapped to 23, 10, and 13, respectively (Appendix A).

## 4. Discussion

*I. lactea* var. *chinensis* is a perennial halophyte. Due to its salt tolerance, drought tolerance, and high ornamental value, *I. lactea* var. *chinensis* has become an important plant for improving saline-alkali land [32,33]. In recent years, our research team has used second-generation sequencing technology to identify genes in *I. lactea* var. *chinensis* related to cadmium and salt stress [10,11], but we did not conduct a comparison of regulatory mechanisms under drought and salt stress. In addition, according to our previous research, *I. lactea* var. *chinensis* is most sensitive under 1.5% NaCl stress [34]. This concentration may cause salt shock in plants and trigger the expression of salt shock-related genes [35]. Therefore, in our study, 1.5% NaCl was used as the treatment concentration to explore the gene expression of *I. lactea* var. *chinensis* under salt shock, and to compare with the treatment of 30% PEG, so as to explore the common regulatory mechanism under salt and drought stress.

Compared with second-generation sequencing technology, SMRT sequencing technology can generate full-length transcripts due to its long reading length. This can greatly improve the accuracy of transcriptome data characterization, which plays an important role in the research of important functional genes in plants [36]. In this study, we used the PacBio RSII platform to analyze expression patterns under NaCl and PEG stress for the first time. The average length of the transcripts using the PacBio RS II platform was 1564 bp, which was 1.97 times higher than the average length of the 793 NGS platform [10]. Overall, 28,419 (91.10%) full-length transcripts were annotated at least once by seven databases, among which 12,556 were annotated in KEGG, 22,089 in COG, 28,063 in NR, 21,697 in GO, and the remaining non-annotated transcripts may represent specific genes. The results will expand the pool of tools available for potential gene exploration. We also used the Illumina platform to analyze the gene expression under NaCl and PEG stress. A total of 62,718 transcripts were identified in nine sample libraries, with more than 20,000 genes in each library. This indicated that the expression of most genes was relatively stable, and the specific expression of genes during abiotic stress was lower. We also found that 3863 DEGs were up-regulated or down-regulated under both NaCl and PEG stress, which indicated that these genes were related to both salt tolerance and drought resistance. These genes will become the focus of future research.

To identify the key genes in response to drought and salt stress further, we carried out WGCNA analysis with all DEGs and selected 10 key genes for salt and drought stress, respectively. According to the results of the NCBI blast, these genes play important roles in plant response to salt stress. For example, *AtOSCA1* in *Arabidopsis thaliana* (*A. thaliana*) could act as a Ca^2+^ permeable channel, which played a key role in the initial stage of the hypertonic stress response [37]. The *PP2C-a10* gene of the PP2C family in wheat could improve the drought resistance of transgenic *A. thaliana* [38]. Ectopic expression of the *TMKP1* protein of the MAPK family in wheat and the *Arabidopsis mkp1* mutant could improve salt stress tolerance [39]. The overexpression of the *OsCPK21* gene could significantly improve the salt tolerance of rice [40], the overexpression of the *PYL5* gene in rice can improve tolerance to drought and salt stress [41], and the overexpression of the *accbl1* gene in the pineapple could significantly improve tolerance to drought and salt stress [42]. According to the analysis results of these genes in other species, we found that these key genes may also play an important role in response to drought and salt stress in *I. lactea* var. *chinensis*, and will also be included in follow-up stress research.

Drought and salt can induce osmotic stress in plants, and then secondary messengers are synthesized [43]. The changes in secondary messengers act as signals to activate different signal cascades. They are the main regulators of signal transduction, which can effectively link external stimuli with cell responses and thus activate the expression of downstream functional genes [3]. In this study, we found 35 DEGs related to osmotic stress signal transduction in DEGs under drought and salt stress, including 6 *CDPK*s, 12 *CIPK*s, and 6 *MAPK*s. Among them, three CDPK-related genes, four CIPK-related genes, and three MAPK-related genes were upregulated, which indicated that the genes related to osmotic stress in *I. lactea* var. *chinensis* would be highly activated within 24 h. Through GO enrichment analysis, it was found that transporter activity, transmembrane transporter activity, and membrane, which are closely related to osmotic stress, were significantly enriched under the two stresses. In addition, 3863 DEGs with the same expression pattern under the two stresses were also significantly enriched in oxidoreductase activity, transmembrane transporter activity, and transmembrane. At the same time, DEGs with the same expression pattern were also enriched in amino acid metabolism, carbohydrate metabolism, signaling and cellular processes, ion channels, and other KEGG pathways, which suggests that there is an obvious co-regulatory system between NaCl and PEG induced osmotic response pathways. These findings are similar to those previously observed in *Arabidopsis* [44], alfalfa [1], rice [45], and red bean [13]. These results showed that when *I. lactea* var. *chinensis* was challenged by drought and salt stress, osmotic stress would occur. At this time, amino acid metabolism, antioxidant enzymes, and other defense systems would be activated through ion channels and signal transduction systems, thus playing a role in response to stress.

The ABA signal transduction pathway is also the core of plant response to abiotic stress. When plants detect abiotic stress, they activate the *SnRK2* receptor through Ca^2+^ signaling, thus stimulating the production of ABA. The *PYL* receptor then binds to ABA and prevents *PP2C*-mediated dephosphorylation of *SnRK2*, resulting in SnRK2 kinase activation. When SnRK2 is activated, it can phosphorylate downstream TFs and activate the expression of ABA-dependent genes, resulting in an ABA response [3]. In this study, we found 22 DEGs related to ABA signal transduction in common DEGs, including six *PYL*, seven *PP2C*, and five *SnRK2* genes. The transcripts of the six *PYL*s were downregulated, while the other genes, including seven *PP2C* and five *SnRK2*, were upregulated after stress. These results were consistent with known ABA regulatory pathways [1,46], implying that abiotic stress may cause the activation of the ABA signaling pathway and affect the expression of *PYL*, *PP2C*, and *SnRK2* genes. *PYR*/*PYL*s were downregulated under salt and drought treatment, while the *PP2C* in the same pathway was induced accordingly.

TFs are important parts of the abiotic stress-mediated signaling pathway, which are phosphorylated by protein kinases, and directly regulate the expression of downstream stress response genes [47]. The GRAS TF family is related to various biological processes, such as root and meristem development, light signal transduction, and biological and abiotic stress responses [48]. Under the two stresses in this study, nine GRAS-related DEGs were co-expressed, and all of them had a trend of upregulated expression, which indicated that *I. lactea* var. *chinensis* could reduce the damage caused by drought and salt stress by regulating the expression of *GRAS*. The MYB family is another type of TF involved in plant responses to environmental stress [49]. Genetic analysis showed that the *TaMYBsm3* gene was overexpressed in wheat, which improved drought tolerance, increased proline content, and decreased MDA content [50]. In this study, 11 MYB-related DEGs were found to be expressed under two stresses, of which six were upregulated, and five were down-regulated. Different expression patterns indicated that the MYB TF family played multiple roles in response to stress. In addition, the NAC and WRKY TF families also play a very important role in the plant response to stress. For example, NAC transcription factor *JUNGBRUNNEN1* in tomato increased drought tolerance [51], and the WRKY TF *GmWRKY12* endowed soybean with drought and salt tolerance [52]. In the two types of DEGs, 23 *NAC*s and 17 *WRKY*s were identified, and all of them were upregulated, which indicated that *I. lactea* var. *chinensis* could protect against drought and salt stress by up-regulating *NAC* and *WRKY* genes. BHLH and AP2/ERF are also important families of TFs in plant response to stress [53,54]. In this study, 6 *bHLHs* and 18 AP2/ERF-related DEGs were found to participate in response to drought and salt stress and showed inducible and inhibitory expression patterns, which indicated that complex transcriptional regulation could be involved in the adaptation of *I. lactea* var. *chinensis* to adverse environments.

Under abiotic stress, a large number of ROS are produced, leading to cell oxidative damage and death. Plants will activate different defense/metabolism pathways to remove and maintain the level of ROS, thus protecting cells from oxidative damage [55]. Previous studies found that 130 DEGs related to active oxygen scavenging were activated in *I. lactea* var. *chinensis* under salt stress, which indicated that *I. lactea* var. *chinensis* could maintain the ROS in cells at a relatively low level by activating an antioxidant defense system [11]. In this study, DEGs shared by *I. lactea* var. *chinensis* under the two stresses were also significantly enriched in GO terms, such as oxidoreductase activity, and KEGG pathways, such as alanine, aspartate, and glutamate metabolism, ascorbate and aldarate metabolism, which indicated that NaCl and PEG stress also activated the antioxidant defense system at the molecular level. In addition, antioxidant enzyme-related DEGs (e.g., *POD*, *CAT*) and non-enzyme antioxidant-related DEGs (e.g., *GST*) were also significantly regulated by the two stresses, which was consistent with previous reports on *I. lactea* var. *chinensis* under salt stress [11]. These results indicate that these DEGs play an important role in response to osmotic stress, thus protecting plants from ROS damage.

## 5. Conclusions

In this study, we sequenced the full-length transcriptome of *I. lactea* var. *chinensis* and analyzed the differential gene expression patterns under NaCl and PEG stress by second-generation sequencing technology. Through WGCNA analysis, key genes were screened in response to stress. Moreover, genes with the same expression pattern were the focus of analysis. Plants are often affected by multiple abiotic stresses. We hope that our research can provide some reference for the cross-study of salt and drought tolerance.

## Figures and Tables

**Figure 1 genes-12-00434-f001:**
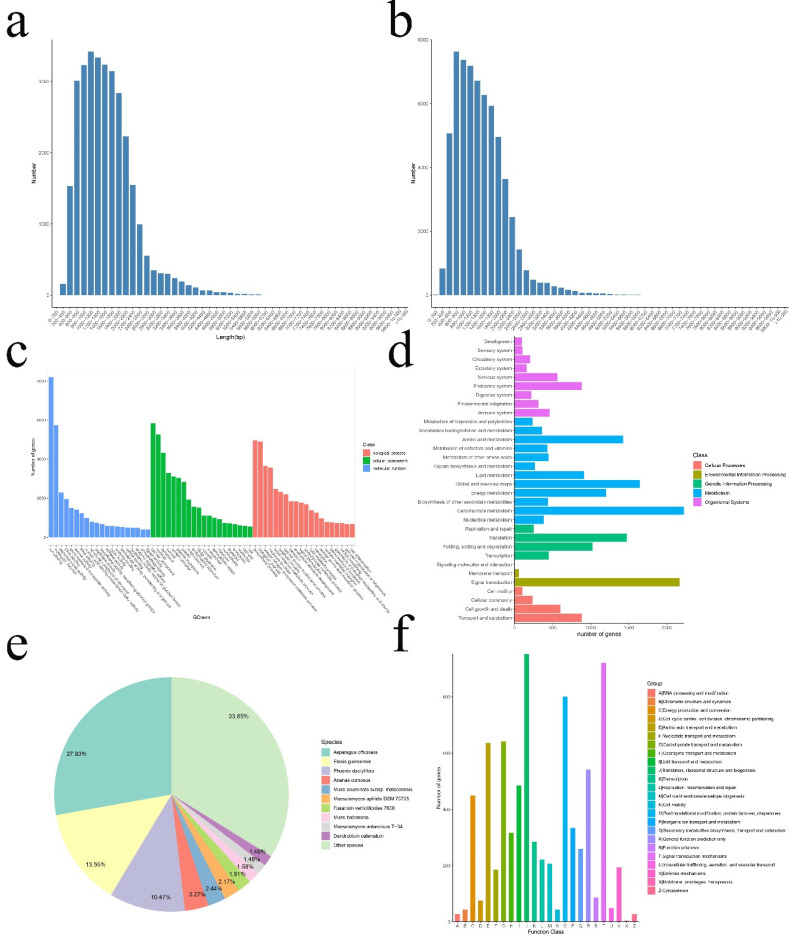
Iso-Seq sequencing, assembly, and annotation of *I. lactea* var. *chinensis*. (**a**) Length distribution of the isoform. (**b**) Length distribution of the full-length transcripts. (**c**) Gene Ontology (GO) classification of the assembled full-length transcripts. (**d**) Kyoto Encyclopedia of Genes and Genomes (KEGG) annotation of the assembled full-length transcripts. (**e**) The distribution of homologous species annotated in the NCBI non-redundant protein (NR) database. (**f**) Cluster of Orthologous Groups of proteins (COG) classification of the assembled full-length transcripts.

**Figure 2 genes-12-00434-f002:**
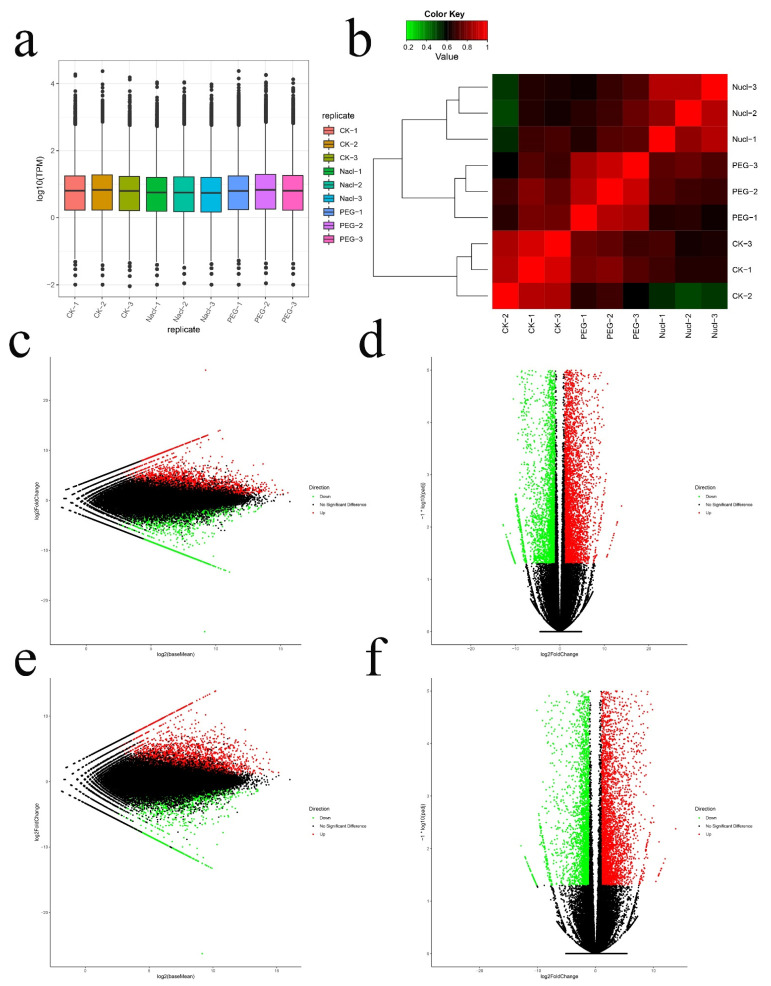
(**a**) The transcript per million (TPM) box plots. (**b**) Clustering analysis diagram between samples. (**c**) CK_VS_NaCl differentially expressed genes (DEGs) M-versus-A (MA) plot. (**d**) CK_VS_NaCl DEGs volcanic plot. (**e**) CK_VS_PEG DEGs MA plot. (**f**) CK_VS_PEG DEGs volcanic plot.

**Figure 3 genes-12-00434-f003:**
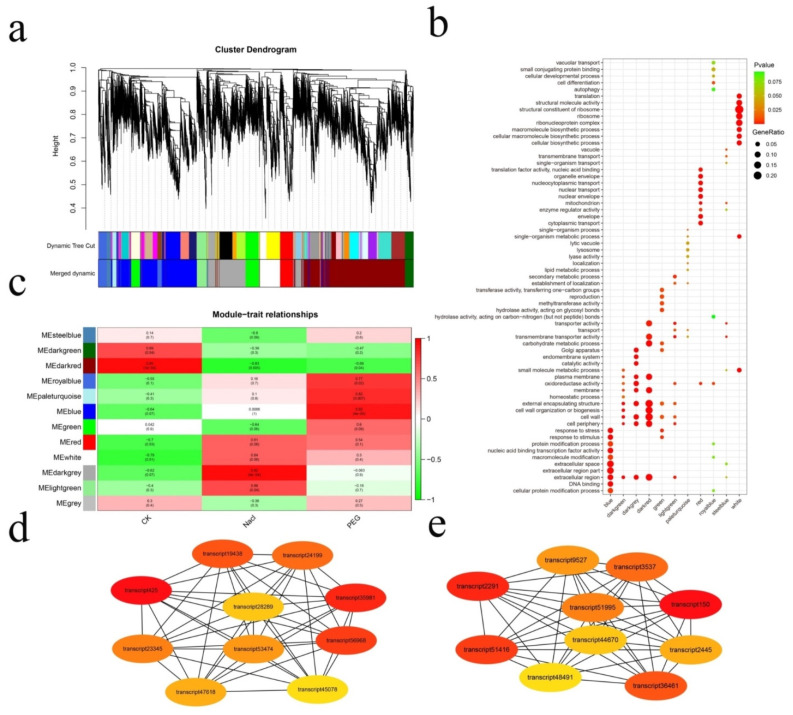
Construction of the gene co-expression network through weighted gene co-expression network analyses (WGCNA). (**a**) Gene dendrogram obtained by hierarchical clustering with the module color. A total of 12 distinct modules were identified. (**b**) Scatterplot of GO enrichment for different modules. The size and color of the dots represent the gene ratio and the range of the P-value, respectively. (**c**) Relationships of modules and different samples. Each row in the table corresponds to a module, and each column corresponds to a sample. (**d**) The 10 hub genes under NaCl stress. (**e**) The 10 hub genes under polyethylene glycol (PEG) stress.

**Figure 4 genes-12-00434-f004:**
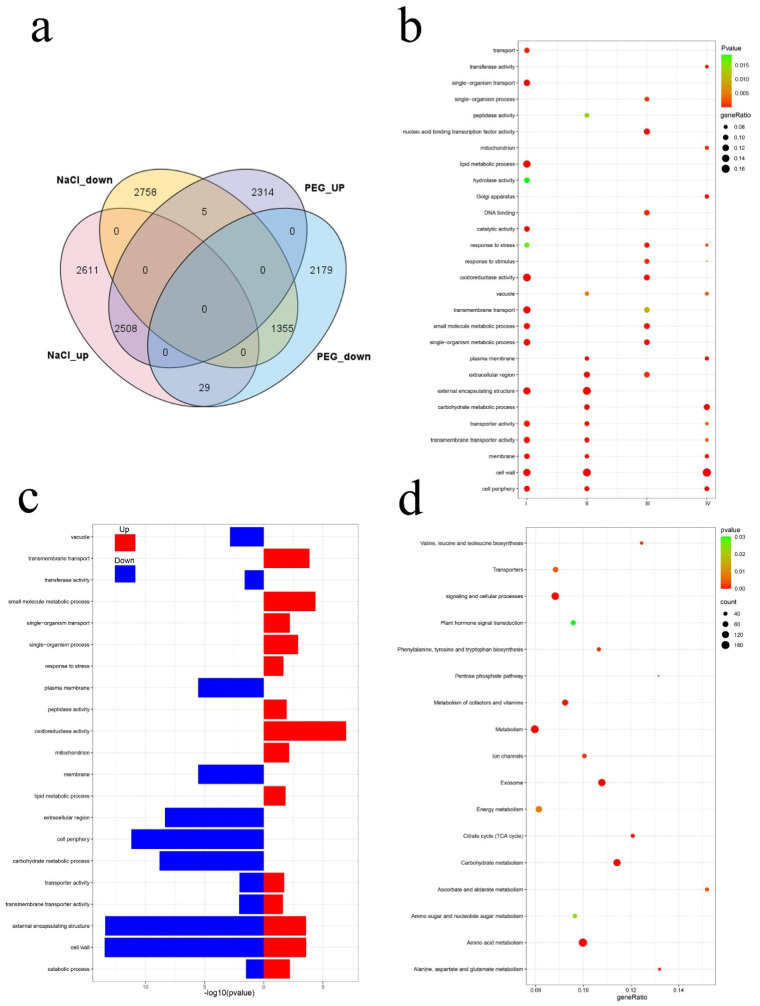
Comparative analysis of the *I. lactea* var. *chinensis* transcriptomes. (**a**) Venn diagram of all DEGs. (**b**) GO enrichment of DEGs that were differentially expressed between control and treatments, the size and color of the dots represent the gene ratio and the range of the *p*-value, respectively. (**c**) GO enrichment of DEGs that had the same expression between NaCl and PEG stress. The degree of GO enrichment is represented by the -log_10_ (*p*-value). (**d**) Scatterplot of enriched KEGG pathways for DEGs that had the same expression under NaCl and PEG stress. The rich factor is the ratio of the DEG number to the total gene number in a certain pathway. The size and color of the dots represent the gene number and the range of the -log_10_ (*q*-value), respectively.

**Figure 5 genes-12-00434-f005:**
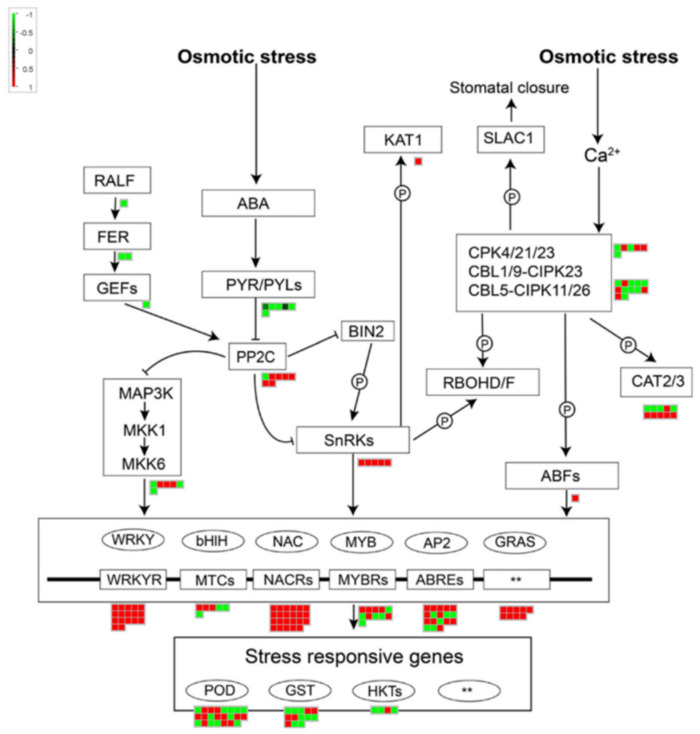
Genes involved in response to salt and drought stress in *I. lactea* var. *chinensis* transcriptomes. Relative expression profiles were showed in green-black-red scale, red indicates log_2_(fold change) > 0, green indicates log_2_(fold change) < 0. ABA, abscisic acid; PYR/PYLs, pyrabactin resistance 1-like protein; PP2C, type 2C protein phosphatases; SnRKs, *Snf1* (sucrose non-fermenting-1)-related protein kinases; BIN2, GSK3-like kinase; RBOHD/F, respiratory burst oxidase homolog D; KAT, K^+^ Channel; SLAC1, slow anion channel 1; CPK/CBL-CIPK, calcium-regulated phosphorylation systems; CAT2/3, catalase 2/3; RALF, rapid alkalinization factor; FER, Feronia; GEF, guanine nucleotide exchange factor; MAPK, mitogen-activated protein kinase; ABFs, ABF, transcription factor; WRKY, WRKY transcription factor; bHlH, bHlH transcription factor; NAC, NAC transcription factor; MYB, MYB transcription factor; AP2, AP2/ERF transcription factor, GRAS, GRAS transcription factor; POD, peroxidase; GST, glutathione S-transferase zeta; HKTs, cation transporters. Adapted with permission from Chen et al. [31]. Copyright 2021 Wiley Online Library, Zhu et al. [3] Copyright 2016 Elsevier, Luo et al. [1] Copyright 2019 Springer Nature. “**” indicate unknown cis-elements.

**Table 1 genes-12-00434-t001:** Summary for the transcriptome data of *I. lactea* var. *chinensis* using PacBio.

	Pacbio
Subreads base (G)	30.89
Subreads number	21,487,251
Average subreads length	1438
CCS	258,615
5′-primer	249,012
3′-primer	249,046
Poly-A	232,150
Flnc	168,924
Average flnc read length	1304
Consensus reads	99,483
Number of isoform	62,717
Number of unigene	31,195
Mean_length	1564
Min_length	161
Max_length	6386
ExN50 (consensus)	1115
ExN90 (consensus)	1693
Number of Genes	31,195
Number of CDS	30,295

## Data Availability

The data have been deposited to the National Center for Biotechnology Information (NCBI) under accession number PRJNA697945.

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
