# Peer review of "Full-Length Transcriptome Sequencing and Comparative Transcriptome Analysis to Evaluate Drought and Salt Stress in Iris lactea var. chinensis"

_genes, 2021, doi:10.3390/genes12030434_

Round 1
Reviewer 1 Report
In their study, Longjie Ni, Zhiquan Wang et al. report on the transcriptome assembly of Iris lactea var. chinensis and its subsequent use to quantify expression changes under two stress conditions. The study design is adequate to address the authors' biological question.
I have attached detailed comments and corrections in an annotated PDF version of your manuscript. Briefly, I have one major concern with the methods and that is the reported use of FPKM with DESeq2. This, if it were done, would be a major analytical flaw. Most likely, this is an improper statement, but the scarcity of the method description and the absence of a public repository for the analysis code impaired my assessment. This is my second most important comment on your work. In accordance to MDPI's policy, but more importantly to ensure a reproducible research that benefits the whole scientific community, making the source code of the analysis publicly available is a must. Ideally, this would be done in a public GitHub repository, which as a matter of fact, can be associated with a DOI and cited in further work. Finally, some attention needs to be given to to the formulation. A few sentences require english editing and the meaning of a few others is obscure.

Author Response
Review 1
Open Review
( ) I would not like to sign my review report
(x) I would like to sign my review report
English language and style
( ) Extensive editing of English language and style required
(x) Moderate English changes required
( ) English language and style are fine/minor spell check required
( ) I don't feel qualified to judge about the English language and style
|
|
Yes |
Can be improved |
Must be improved |
Not applicable |
|
Does the introduction provide sufficient background and include all relevant references? |
(x) |
( ) |
( ) |
( ) |
|
Is the research design appropriate? |
( ) |
( ) |
(x) |
( ) |
|
Are the methods adequately described? |
( ) |
( ) |
(x) |
( ) |
|
Are the results clearly presented? |
( ) |
(x) |
( ) |
( ) |
|
Are the conclusions supported by the results? |
( ) |
(x) |
( ) |
( ) |
Comments and Suggestions for Authors
In their study, Longjie Ni, Zhiquan Wang et al. report on the transcriptome assembly of Iris lactea var. Chinensis and its subsequent use to quantify expression changes under two stress conditions. The study design is adequate to address the authors' biological question.
I have attached detailed comments and corrections in an annotated PDF version of your manuscript. Briefly, I have one major concern with the methods and that is the reported use of FPKM with DESeq2. This, if it were done, would be a major analytical flaw. Most likely, this is an improper statement, but the scarcity of the method description and the absence of a public repository for the analysis code impaired my assessment. This is my second most important comment on your work. In accordance to MDPI's policy, but more importantly to ensure a reproducible research that benefits the whole scientific community, making the source code of the analysis publicly available is a must. Ideally, this would be done in a public GitHub repository, which as a matter of fact, can be associated with a DOI and cited in further work. Finally, some attention needs to be given to to the formulation. A few sentences require english editing and the meaning of a few others is obscure.
Answer: Thank you very much for useful comments of our manuscript. We have made careful revisions, and the detailed corrections are listed below point by point. The revised manuscript was reviewed and corrected the numerous grammatical errors by the International Science Editing’s Language Editing service (http://www.internationalscienceediting.com/) before submission. Please let me know if you have any question!
Point 1: Third generation has a worse accuracy, an average 10% error rate vs. Illumina’s 1%. Correct that sentence and your meaning.
Response 1: Thank you for your comments! I am sorry that I made the mistake here, and we have corrected the sentence in the revised manuscript (line 73).
Point 2: This is also nowadays incorrect, with the latest IsoSeq CCS data being of higher quality than Illumina’s reads. As such correction by Illumina is not recommended anymore. However, hybrid assemblies are still useful because of the lack of IsoSeq sequencing depth. And Illumina is still required for quantification. Adapt your sentence accordingly.
Response 2: Thank you for your useful comments! We have adapted the sentences in the revised manuscript (lines 71-75).
Point 3:What concentration and volume?
Response 3: Thank you for your comments! The concentration of RNA of each sample was greater than 300 ng/ul, and the total amount of RNA used for cDNA library was greater than 5 ug (lines 115-118 in the revised manuscript).
Point 4: sequenced on a PacBio sequel II instrument.
Response 4: Thank you for your comments! We have made relative revision (lines 122-123).
Point 5: This is not suffcient, detail which tools were used for doing that, the exact pipeline, the tools version, the non default arguments, etc. Anything needed to reproduce the assembly from the raw data.
Response 5: Thank you for your comments! We have added more details about tool information and analysis steps improved in the revised manuscript (lines 113-145).
Point 6:clarify 3) and 4), they make no sense to me at present.
Response 6:Thank you for your comments! As 3) and 4) were default parameters, we have deleted then in the revised manuscript (lines 135-137).
Point 7: Same as above, all tools, versions, parameters should be given. Any ad- hoc scripts should be made available in either the supplement or a public GitHub repository.
Response 7: Thank you for your comments! We have made relative revision (lines 138-145).
Point 8: RSEM is acceptable, but there are better alternatives. Consider salmon for future work. RSEM reports TPM and FPKM should never be used as they are flawed. There is enough evidence of that in the literature. See this for some pointers: https://docs.google.com/document/d/1D5CoNPxy45MpXLLvbIImFzCebFfe0jD5KLeAQibGTI, If you used FPKM for any analysis, these would need redoing either with RSEM TPM or better using CPM from edgeR that correct for the library size factor (difference in sequencing depth) better.
Response 8: Thank you for your comments! We have analyzed the FPKM with edgeR and adapted relative results in the revised manuscript (lines 179-183).
Point 9: If this is reallty what was done, using FPKM for DESeq2, then all analyses are worthless and need to be redone. Look at the DESeq2 documentation accurately, where it is stated in bold that raw counts should be used and nothing else.
Response 9: Thank you for your comments! I am so sorry that I had made a mistake in the statement here, and we have corrected the mistake in the modified version (lines 181-183).
Point 10: Justify the choice for the cutoff on log2FC and FDR. Check Schurch et al., RNA, 2016.
Response 10: Thank you for your comments! We have added the reason for choosing the cutoff on log2FC and FDR based on previous studies (Check Schurch et al.,2016) in the revised manuscript (line 183).
Schurch NJ, Schofield P, Gierliński M, Cole C, Sherstnev A, Singh V, Wrobel N, Gharbi K, Simpson GG, Owen-Hughes T, Blaxter M, Barton GJ. How many biological replicates are needed in an RNA-seq experiment and which differential expression tool should you use? RNA. 2016 Jun;22(6):839-51. doi: 10.1261/rna.053959.115.
Point 11: I do not understand what the preference of DEG length means.
Response 11: Thank you for your comments! And I am sorry I did not explain it clearly, DEG length is gene length bias in DEGs based on previous studies (Young et al., 2010), we have made explanation in the revised manuscript (line 202).
Young, M.D., Wakefield, M.J., Smyth, G.K., Oshlack, A. Gene ontology analysis for RNA-seq: accounting for selection bias. Genome Biol 2010, 11, R14, doi:10.1186/gb-2010-11-2-r14.
Point 12: Provide all R scripts for reproducibility. See my previous comments.
Response 12: Thank you for your comments! We have provided all R scripts (lines 200-208 in the revised version).
Point 13: N50 is a useless metric for transcript assembly. Consider reporting the ExN50 or ExN90 instead: https://github.com/trinityrnaseq/trinityrnaseq/wiki/Transcriptome-Contig-Nx-and-ExN50-stats Actually, here reporting on the BUSCO score for your assembly would be by far more informative
Response 13: Thank you for your comments! We have recalculated the values of EXN50 and EXN90 and made the changes in Table 1. And the future study is necessary to validate the BUSCO score, but technical imperfection is a difficult problem for reporting now, we have added the NR and COG database annotation information in the revised manuscript (lines 235-244) to provide more information about Iso-Seq sequencing, assembly, and annotation. If experts and editors feel that BUSCO is essential, please review to us again, we will try my best to perform it.
Point 14:What of the 900 sequences without CDS? What are they?
Response 14: Thank you for your comments! And I am sorry I did not explain it clearly, 900 sequences without CDS, which may be pseudogenes and lncRNAs (Furuno M.,2003). And we have added relative information in the revised manuscript (lines 226-227).
Furuno M, Kasukawa T, Saito R, Adachi J, Suzuki H, Baldarelli R, Hayashizaki Y, Okazaki Y. CDS annotation in full-length cDNA sequence. Genome Res. 2003 Jun;13(6B):1478-87. doi: 10.1101/gr.1060303. PMID: 12819146; PMCID: PMC403693.
Point 15: Revise the english.
Response 15: Thank you for your comments!We have been revised in the revised version (lines 250-253).
Point 16: Such a validation is useless, remove it. It has been done comprehensively by much larger consortium(SEQC, Nature, 2014) and not by cherry picking a few genes. Instead use that space to extend on the other results and complete the methods. Or report additional more informative results such as that from BUSCO.
Response 16: Thank you for your comments! Due to technical problems, BUSCO has not been validated, and we have added the information of DEGs MA map and volcano map together with the result of qRT-PCR in this space to analyze the expression situation in the revised version (lines 255-260, Fig.2a-d.) If experts and editors feel that BUSCO is essential, please review to us again, we will try my best to perform it.
Point 17: Validate your choice for 13. Adding the method will help clarify that.
Response 17: Thank you for your comments! And I am sorry I did not explain it clearly, we have added the relative method in the revised version (lines 280-282).
Point 18: Under which criteria were the 10 genes selected?
Response 18: Thank you for your comments! I am sorry I did not explain it clearly, we used Cytoscape Plugin Cytohubba for selecting hubgene based on previous studies (Chin et al., 2014), and we have explained this point in the revised manuscript (lines 208 and lines 308-311).
Chin, C.-H.; Chen, S.-H.; Wu, H.-H.; Ho, C.-W.; Ko, M.-T.; Lin, C.-Y. cytoHubba: identifying hub objects and sub-networks from complex interactome. BMC Syst Biol 2014, 8 Suppl 4, S11, doi:10.1186/1752-0509-8-S4-S11.
Point 19: I suppose NCBI nr? How was the annotation done?
Response 19: Thank you for your comments! I am sorry I did not explain it clearly, we used BLAST for annotations in NCBI and the details were showed in the revised manuscript (lines 310-311).
Point 20: A central assumption to DE analysis using DESeq2 is that the vast majority of the genes are not differentially expressed. Showing volcano plots, MA plots for both stress would be necessary to allow the reader assessing that.
Response 20: Thank you for your comments! We have added volcano plots and MA plots in Fig. 2a-d in the revised version.
Point 21: What justifies separating up and down DE genes? Why can’t genes be oppositely affected under either stress? Show a Venn diagram including both.
Response 21: Thank you for your comments! I am sorry I did not explain it clearly, 34 DEGs were found having different expression patterns under the two stresses. We have re-made a Venn diagram to demonstrate DEGs of different expression modes in Fig. 4a in the revised version and added relative states (lines 335-336).
Point 22: Again, there is biological rationale for that. Genes will be part of pathways, most of which include (negative) feedback loops, where the up-regulation of a gene results in the down-regulation of other members of the same pathway. Exactly as you show in Fig.5. Extend your analysis taking this into consideration.
Response 22: Thank you for your comments! I am sorry I did not state this issue clearly. In our study, we found genes in the same pathway showing different expression trends as shown in Fig. 4 and Fig. 5. For example, PYR/PYLs were down-regulated under salt and drought treatment, while the PP2C in the same pathway were induced accordingly, which is according with previous studies (Luo, D.,2019). We have added relative content in the revised version (lines 339-342 and lines 489-497 ).
Luo, D.; Zhou, Q.; Wu, Y.; Chai, X.; Liu, W.; Wang, Y.; Yang, Q.; Wang, Z.; Liu, Z. Full-length transcript sequencing and comparative transcriptomic analysis to evaluate the contribution of osmotic and ionic stress components towards salinity tolerance in the roots of cultivated alfalfa (Medicago sativa L.). BMC Plant Biology 2019, 19, doi:10.1186/s12870-019-1630-4.
Point 23: This is incoherent throughout the MS. Did you use RSII or sequel (2)?
Response 23: Thank you for your comments! I am sorry I made a mistake, and we used RSII in this study and we have revised (lines 430).
Point 24: Here, you should hypothesise on how many genes/transcripts may have been missed by your approach of using PacBio, due to its low sequencing depth. This could easily be estimated by running a de-novo transcript assembly of the RNA-Seq data using trinity and checking how many trinity transcripts have no PacBio support. This may be biased by chimeric transcripts, but testing this on the ExN90 subset would be informative.
Response 24: Thank you for your comments! I am sorry that I did not state this issue clearly. The average of total mapping rates of reads obtained by RNA-Seq with PacBio was higher than 75%, so the few genes/transcripts were missed by PacBio. We have added relative information in the revised manuscript (lines 250-254).
Point 25: How does an average of ~20,000 genes with a total of > 60,000 genes show stability? Report more accurate metrics to support that claim.
Response 25: Thank you for your comments! I am sorry I had made a mistake in the statement here, 62,718 transcript were identified in total 9 sample libraries, with an average of more than 19,250 genes in each library. I have corrected the mistake in the modified version (line 438-439).
Point 26: You do not present any evidence of co-expression. Rather you selected genes that are DE in both experiments. This is vastly different.
Response 26: Thank you for your comments! I am so sorry that I had made a mistake in the statement, it should be stated in this way “3863 DEGs with the same expression pattern under the two stresses” not “co-expression”. We have corrected the mistake in the modified version (line 472-474).
Point 27: Why are the numbers different? Also be careful throughout the manuscript whether you use genes or transcripts. If talking about the results of the assembly, these are transcripts If talking about the clustered assembly results, these could be equated to genes. Correct throughout.
Response 27: Thank you for your comments! I am so sorry that I had made a mistake and I have corrected the mistake throughout in the modified version (lines 489-497).

Reviewer 2 Report
Authors used Pac Bio third generation sequencing (single molecule real-time sequencing) to obtain information about full-length transcripts of Iris lactea var. chinensis. Unigenes were annotated by BLASTX software using seven databases. Then the Illumina Hi-Seq 2000 platform was used to quantitatively assess the expression level of genes after NaCl and PEG treatment. Data of differential gene expression were validated by qRT-PCR.
Authors analysed also coexpression networks of differentially expressed genes to group them into 13 differential group modules. Based on enrichment of different modules Authors identified 10 genes that are play a key role in response to salt stress and drought stress.
The work is new, original and add valuable data to plant transcriptomic studies of I. lactea.
In my opinion the content of material and methods section should be improved to assure the correctness of obtained results.
In my opinion the generally valuable article could be improved, mainly by adding some details to the Material and Method or Discussion Section. Following changes should be included:
1.Paragraph 2.1
In the part concerning induction of salt stress conditions Authors used 1.5% NaCl. It corresponds to about 250 mM NaCl. According to a data presented in review of Shavrukov 2013 (data below) and other experimental works, such concentrations of NaCl applied instantly, not stepwise to plant tissue can cause rather salt shock, not salt stress.
I think that in the discussion section Authors could discuss in several sentences basic definitions of salt stress and salt shock (for example according to Shavrukov 2013 or other works) and differentiate among them.
Shavrukov Y “Salt stress or salt shock: which genes are we studying?” J Exp Bot, vol. 64,issue 1, 119-127pp, 2013.
- Paragraph 2.2, 2.3
The length of sequences produced by PacBio and Oxford Nanopore Technology is longer than NGS. However a common problem is a relatively high error rates. Authors should describe method/strategy used for error correction.
Name of software used for grouping, clustering and correcting.
- Paragraph 2.4 qRT-PCR analysis
If possible Authors should add information concerning the RNA integrity- for example based on Agilent Bioanalyzer 2100 tests
Details of RT reaction: amount of RNA and RT reaction volume, details of RT reaction-temperature, time, used enzyme name (reverse transcriptase) and concentration, enzyme or kit supplier, storage conditions of obtained cDNA.
Citation of the used 2-ΔΔCt method should be added.
Software used to analyse qPCR data should be described.
Author Response
Review 2
Open Review
(x) I would not like to sign my review report
( ) I would like to sign my review report
English language and style
( ) Extensive editing of English language and style required
( ) Moderate English changes required
(x) English language and style are fine/minor spell check required
( ) I don't feel qualified to judge about the English language and style
|
|
|
|
Yes |
Can be improved |
Must be improved |
Not applicable |
|
Does the introduction provide sufficient background and include all relevant references? |
(x) |
( ) |
( ) |
( ) |
|
Is the research design appropriate? |
( ) |
(x) |
( ) |
( ) |
|
Are the methods adequately described? |
( ) |
(x) |
( ) |
( ) |
|
Are the results clearly presented? |
(x) |
( ) |
( ) |
( ) |
|
Are the conclusions supported by the results? |
(x) |
( ) |
( ) |
( ) |
Comments and Suggestions for Authors
Authors used Pac Bio third generation sequencing (single molecule real-time sequencing) to obtain information about full-length transcripts of Iris lactea var. chinensis. Unigenes were annotated by BLASTX software using seven databases. Then the Illumina Hi-Seq 2000 platform was used to quantitatively assess the expression level of genes after NaCl and PEG treatment. Data of differential gene expression were validated by qRT-PCR.
Authors analysed also coexpression networks of differentially expressed genes to group them into 13 differential group modules. Based on enrichment of different modules Authors identified 10 genes that are play a key role in response to salt stress and drought stress.
The work is new, original and add valuable data to plant transcriptomic studies of I. lactea.
In my opinion the content of material and methods section should be improved to assure the correctness of obtained results.
In my opinion the generally valuable article could be improved, mainly by adding some details to the Material and Method or Discussion Section. Following changes should be included:
Answer: Thank you very much for meaningful comments of our manuscript. We have made careful revisions and hope that the correction will meet with approval. The detailed corrections are listed below point by point. Please let me know if you have any question!
Point 1: In the part concerning induction of salt stress conditions Authors used 1.5% NaCl. It corresponds to about 250 mM NaCl. According to a data presented in review of Shavrukov 2013 (data below) and other experimental works, such concentrations of NaCl applied instantly, not stepwise to plant tissue can cause rather salt shock, not salt stress.
I think that in the discussion section Authors could discuss in several sentences basic definitions of salt stress and salt shock (for example according to Shavrukov 2013 or other works) and differentiate among them.
Shavrukov Y “Salt stress or salt shock: which genes are we studying?” J Exp Bot, vol. 64,issue 1, 119-127pp, 2013.
Response 1: Thank you for your comments! We have discussed the basic definitions of salt stress and salt shock and also added the reason for choosing the concentration, the details were showed in the revisd version (lines 419-425).
Point 2: The length of sequences produced by PacBio and Oxford Nanopore Technology is longer than NGS. However a common problem is a relatively high error rates. Authors should describe method/strategy used for error correction. Name of software used for grouping, clustering and correcting.
Response 2: Thank you for your comments! We have described method/strategy used for error correction in the revised manuscript (lines 103-208).
Point 3: Paragraph 2.4 qRT-PCR analysis
If possible Authors should add information concerning the RNA integrity- for example based on Agilent Bioanalyzer 2100 tests.
Details of RT reaction: amount of RNA and RT reaction volume, details of RT reaction-temperature, time, used enzyme name (reverse transcriptase) and concentration, enzyme or kit supplier, storage conditions of obtained cDNA.
Citation of the used 2-ΔΔCt method should be added.
Software used to analyse qPCR data should be described.
Response 3:Thank you for your comments! We have added relative information about the analysis of qRT-PCR, the details were showed in the revised manuscript (lines 186-189 and 196-198).

Round 2
Reviewer 1 Report
In their revised manuscript, Longjie Ni, Zhiquan Wang et al. have addressed my previous concerns but only partially so. The greater information added to the methods has helped me perform a better assessment of the methodology used. There remains a number of unclarities, some of my previous comments were not addressed and the methodology used for the DE analysis is flawed. See the annotated manuscript for further details and examples of some of the points listed below.
1) The qRT-PCR validation is not necessary and should be removed. Using 9 cherry-picked genes has zero statistical power to assed the quality of your transcriptome.
2) One should not use FPKM, but if at all, only for graphical representation. RSEM provides TPM, use these instead as they are slightly more robust than FPKM.
3) As I mentioned earlier, using anything else than raw counts for the DE analysis with edgeR or DESeq2 is FLAWED! Reading the authors answer and the revised methods, FPKM were used, further normalised with edgeR (how so?) and then analysed either with DEGseq or DESeq2 for differential expression (the methods mentions DEGseq2 and cite DEGseq while DESeq2 is mentioned in the results). The results of such a pipeline are FLAWED.
4) While I would appreciate seeing the results from BUSCO as they are highly informative, there is by now evidence that the assembly is of good quality. Nonetheless, running BUSCO and additional assessment such as detonate would strengthen that claim and make your readership more confident of the results shown. These would be much more relevant than the qRT-PCR.
5) I do not find any reference to the code developed for the different analyses, neither is there a reference to a github repository nor is it in the supplementary file. Please provide it. As I wrote earlier, it is to be "in accordance" with "MDPI's policy, but more importantly to ensure a reproducible research that benefits the whole scientific community, making the source code of the analysis publicly available is a must. Ideally, this would be done in a public GitHub repository, which as a matter of fact, can be associated with a DOI and cited in further work."
6) The use of professional english editing helped clarify the text, but some of the new additions would need some attention to be given to the formulation, grammar, time concordance, etc.
Author Response
Review 1
Comments and Suggestions for Authors
In their revised manuscript, Longjie Ni, Zhiquan Wang et al. have addressed my previous concerns but only partially so. The greater information added to the methods has helped me perform a better assessment of the methodology used. There remains a number of unclarities, some of my previous comments were not addressed and the methodology used for the DE analysis is flawed. See the annotated manuscript for further details and examples of some of the points listed below.
Answer: Thank you very much for useful comments of our manuscript! We have made careful revisions, and the detailed corrections are listed below point by point. Please let me know if you have any question!
Point 1: The qRT-PCR validation is not necessary and should be removed. Using 9 cherry-picked genes has zero statistical power to assed the quality of your transcriptome.
Response 1: Thank you for your comments! We have removed the content of qRT-PCR validation and added the results of BUSCO in the revised version (lines129-131 and lines 187-188).
Point 2: One should not use FPKM, but if at all, only for graphical representation. RSEM provides TPM, use these instead as they are slightly more robust than FPKM.
Response 2: Thank you for your comments! We have converted FPKM values into TPM based on previous study (Wagner et al., 2012) and used them for subsequent WGCNA analysis. The details were showed in the revised version (lines 153-156, lines 227-262 and lines 377-385).
Wagner GP, Kin K, Lynch VJ. Measurement of mRNA abundance using RNA-seq data: RPKM measure is inconsistent among samples. Theory Biosci. 2012 Dec;131(4):281-5. doi: 10.1007/s12064-012-0162-3. Epub 2012 Aug 8. PMID: 22872506
Point 3: As I mentioned earlier, using anything else than raw counts for the DE analysis with edgeR or DESeq2 is FLAWED! Reading the authors answer and the revised methods, FPKM were used, further normalised with edgeR (how so?) and then analysed either with DEGseq or DESeq2 for differential expression (the methods mentions DEGseq2 and cite DEGseq while DESeq2 is mentioned in the results). The results of such a pipeline are FLAWED.
Response 3: Thank you for your comments! I am so sorry that I had made a mistake in the statement here, we used the raw read counts for analysis with software, not FPKM, and we use edgeR and DEseq2 software for DEGs analysis (Love et al., 2014). We have made relative revision and corresponding references are also inserted (lines 158-162 and lines 217).
Love, M.I., Huber, W. & Anders, S. Moderated estimation of fold change and dispersion for RNA-seq data with DESeq2. Genome Biol 15, 550 (2014). https://doi.org/10.1186/s13059-014-0550-8
Point 4: While I would appreciate seeing the results from BUSCO as they are highly informative, there is by now evidence that the assembly is of good quality. Nonetheless, running BUSCO and additional assessment such as detonate would strengthen that claim and make your readership more confident of the results shown. These would be much more relevant than the qRT-PCR.
Response 4: Thank you for your comments! We have removed the content of qRT-PCR validation and added the results of BUSCO in the revised version (lines 129-131 and lines 187-188).
Point 5: I do not find any reference to the code developed for the different analyses, neither is there a reference to a github repository nor is it in the supplementary file. Please provide it. As I wrote earlier, it is to be "in accordance" with "MDPI's policy, , but more importantly to ensure a reproducible research that benefits the whole scientific community, making the source code of the analysis publicly available is a must. Ideally, this would be done in a public GitHub repository, which as a matter of fact, can be associated with a DOI and cited in further work."
Response 5: Thank you for your comments! I am so sorry that I had made a mistake here, in this study, we used DESeq2 for the different analyses based on previous study (Love et al., 2014). We have made relative revision (lines 158-162).
Love, M.I., Huber, W. & Anders, S. Moderated estimation of fold change and dispersion for RNA-seq data with DESeq2. Genome Biol 15, 550 (2014). https://doi.org/10.1186/s13059-014-0550-8
Point 6: The use of professional english editing helped clarify the text, but some of the new additions would need some attention to be given to the formulation, grammar, time concordance, etc.
Response 6: Thank you for your comments! We have checked and modified the whole text in the revised version.
